# Polarization Super-Resolution Imaging Method Based on Deep Compressed Sensing

**DOI:** 10.3390/s22249676

**Published:** 2022-12-10

**Authors:** Miao Xu, Chao Wang, Kaikai Wang, Haodong Shi, Yingchao Li, Huilin Jiang

**Affiliations:** School of Opto-Electronic Engineering, Changchun University of Science and Technology, Changchun 130022, China

**Keywords:** super-resolution, deep compressed sensing, polarization imaging

## Abstract

The division of focal plane (DoFP) polarization imaging sensors, which can simultaneously acquire the target’s two-dimensional spatial information and polarization information, improves the detection resolution and recognition capability by capturing the difference in polarization characteristics between the target and the background. In this paper, we propose a novel polarization imaging method based on deep compressed sensing (DCS) by adding digital micromirror devices (DMD) to an optical system and simulating the polarization transmission model of the optical system to reconstruct high-resolution images under low sampling rate conditions. By building a simulated dataset, training a polarization super-resolution imaging network, and showing excellent reconstructions on real shooting scenes, compared to current algorithms, our model has a higher peak signal-to-noise ratio (PSNR), which validates the feasibility of our approach.

## 1. Introduction

Polarization is one of the important physical properties of light. Different targets show different polarization characteristics due to differences in morphology, texture, water content, and dielectric coefficient of targets. Polarization imaging technology integrates spatial, spectral, and polarization information of the target object, which cannot only improve the information for acquiring the target object but also enhance the ability to detect and identify the target object [1]. At the same time, polarization imaging is a new type of photoelectric detection system. By using polarization imaging to measure the objective reflection of the degree of polarization and polarization angle information, the dimension of imaging information is increased, and the polarization information and the light intensity distribution in two-dimensional space are obtained. Currently, polarization imaging techniques have been widely used in the fields of target detection [2], image defogging [3], biomedicine [4], and remote sensing [5].

In recent years, a division of focal plane (DoFP) polarization sensor has been developed, which integrates an array of micro-polarizers of four polarization states into the image element of a focal plane sensor to collect polarization information of four polarization states simultaneously. This sensor can realize polarization imaging in real-time and ensure that each measurement is performed under the same illumination or radiation conditions. Compared with the time-sharing polarization device, it has the advantages of good real-time, small volume, small mass, compact structure, and high integration. However, the DoFP polarization sensor inevitably causes the problem of half the resolution, so improving the resolution while keeping fast polarization imaging remains a major challenge.

In order to solve the polarization resolution reduction problem, various methods have been proposed, such as interpolation and deep learning methods. The interpolation algorithms include bicubic interpolation, Newton polynomial interpolation [6], etc. Deep learning-based reconstruction methods include the polarization image super-resolution reconstruction network PDCNN [7], the end-to-end output of intensity images, line polarization degree images, and polarization angle images of Fork -Net [8] and other methods. These methods mainly focus on the super-resolution reconstruction of polarized images and do not consider the affection of the optical system. With the development of computational imaging technology, computational imaging methods that incorporate the Digital Micromirror Device (DMD) in optical systems to compress and encode the optical path and subsequently reconstruct high-resolution images are widely used in microscopy, medium wave infrared imaging [9], holographic imaging [10], and spectral imaging.

Our technique relies on the scene information being sampled and encoded by the computer-controlled DMD device, then imaged by the DoFP detector, and finally reconstructed by the depth learning network. The polarization aberration model of the optical system designed was established based on the polarization transmission theory, and the training data with polarization aberration were obtained to train the super-resolution reconstruction network. The experimental platform was built to verify the feasibility of the proposed network in real application scenarios, and the high-resolution polarization images were successfully reconstructed with a low sampling rate.

## 2. Materials and Methods

### 2.1. Block Compressed Polarization Imaging

The block-compressed polarization imaging method is an application of the block-compressed imaging (BCI) theory [6] on the field of polarization imaging, combined with the imaging principle of DoFP polarization imaging to obtain low sampling rate and high-resolution images of polarization information. The DoFP micro-polarizer sensor applied is composed of a CMOS photodiode sensor and a layer of polarizer added on the top of the photodiode, as shown in Figure 1. Every four adjacent pixels is used as a computational unit, and each pixel takes the polarized light at only one of four orientations, e.g., 0, 45, 90, and 135 degrees [6].

The light intensity of different polarization transmission angles $I_{0^{\circ }}$, $I_{45^{\circ }}$, $I_{90^{\circ }}$ and $I_{135^{\circ }}$ is obtained by the four image elements. Since the circular polarization component of the natural target is small, the Stokes vector S can be given by
(1)$$S=\left(\begin{array}{c} S_{0}\\ S_{1}\\ S_{2} \end{array}\right)=\left(\begin{array}{c} I_{0^{\circ }}+I_{90^{\circ }}\\ I_{0^{\circ }}-I_{90^{\circ }}\\ I_{45^{\circ }}-I_{135^{\circ }} \end{array}\right)$$

The degree of linear polarization (*DoLP*)and the angle of polarization (*AoP*) are given by
(2)$$DoLP=\frac{\sqrt{S_{1}^{2}+S_{2}^{2}}}{S_{0}}$$
(3)$$AoP=\frac{1}{2}\arctan\frac{S_{2}}{S_{1}}$$

However, this imaging method inevitably brings a reduction in the resolution, with the polarized image resolution reduced to half of the resolution of the original detector. DMD devices are widely used in compressive imaging because of the high resolution, high reflectivity, and quick response [7]. DMD is an essentially reflective digital semiconductor light modulator consisting of millions of micromirrors on a semiconductor silicon substrate [11]. Each micromirror is controlled by electrostatics independently to determine the angle of deflection of the light. BCI imaging methods can use low-resolution detectors paired with high-resolution DMDs to achieve high-resolution imaging results at a lower cost on the detectors, especially in the medium wave infrared band. Based on our previous optical design results for super-resolution imaging systems [8], a DMD-based polarization super-resolution imaging optical system is designed by combining the above-mentioned Micro-polarizer array DoFP detector. The imaging optical system mainly consists of four parts: telescope objective, DMD, projection objective, and DoFP polarization detector. The target scene is imaged by the telescope objective in the intermediate image plane. Then, the intermediate image is modulated by the DMD and finally imaged on the FPA by the projection objective. The main system parameters are shown in Table 1.

The structure of the optical system is shown in Figure 2a, and the modulation transfer function (MTF) of the optical system is shown in Figure 2b. It can be seen that the optical system has good imaging quality with MTF values close to the diffraction limit, basically avoiding the imaging quality degradation in the super-resolution reconstruction introduced by the wave aberration.

Firstly, the target scene is sub-regionally processed by the DMD, each sub-region of the scene is projected onto the DMD corresponding sub-region and encoded simultaneously, and then the encoded light intensity information is captured by the focal plane detector, where each pixel of the detector corresponds to a sub-region of the scene, and then the acquired image information is used to reconstruct of the target scene image by the reconstruction algorithm.

### 2.2. Analysis of the Effect of Polarization Aberration on Imaging

Polarization aberration is used to characterize the change in amplitude, phase, and polarization state of the light after the light traveling through an optical system [12]. The off-axis optical system produces polarization aberration, which affects the polarization imaging quality [13]. To characterize the polarization aberration, a 3 × 3 coherency matrix was used to perform ray tracing of the DMD-based polarization optical system. We transform the local 3D eigen-vibration coordinates into the 3D global coordinates as shown in Figure 3.

Coherency matrix in the eigen-vibration coordinates

The vibrational plane of the vector is called the eigen-vibration plane of the ray. For the linearly polarized light with a light intensity $I_{a}$, its eigen-vibration plane and propagation direction are described by the three-dimensional incoherence matrix $J_{P3L}$, as follows:(4)$$J_{P3L}=\frac{I_{a}}{2}\left(\begin{array}{ccc} 1+DoLP\cos2AoP & DoLP\sin2AoP & 0\\ DoLP\sin2AoP & 1-DoLP\cos2AoP & 0\\ 0 & 0 & 0 \end{array}\right)$$

2.Coherency matrix in global coordinates

To describe the polarization characteristics of the light after passing through different optical interfaces in the same coordinates, it is necessary to transform the coordinates of the light being traced and unify it into the global coordinate system. *R* is the transformation matrix from the eigen coordinate system to the global coordinate system; therefore, the three-dimensional global coherency matrix of the incident ray $J_{P3(\mathrm{in})}$ being traced can be expressed as:(5)$$J_{P3(\mathrm{in})}=R\cdot J_{P3L}\cdot R^{T}$$

3.Coherency matrix in interface coordinates

The three-dimensional global coordinate system is O-XYZ, but the coordinate system of the incident, refracted, and reflected rays is the S-P-K interface coordinate system. In order to describe the polarization transformation of the optical surface in the global coordinate system, it is also necessary to transform the interface coordinate system S-P-K into the global coordinate system. $T_{3r(q)}$ denotes the Jones matrix of the optical interface q in the 3D interface coordinate system. The above transformation is carried out on each surface of the optical system to obtain the three-dimensional polarization tracking matrix $P_{3(q)}$ of each surface;
(6)$$P_{3(q)}=\left[\begin{array}{ccc} p_{\left(out,q\right)} & s_{\left(out,q\right)} & k_{\left(out,q\right)} \end{array}\right]\cdot T_{3r(q)}\cdot {\left[\begin{array}{ccc} p_{\left(in,q\right)} & s_{\left(in,q\right)} & k_{\left(in,q\right)} \end{array}\right]}^{T}$$

The 3D polarization trace matrix of the entire optical system $P_{3\left(opt\right)}$ is obtained by multiplying the trace matrices of all interfaces together.
(7)$$P_{3(\mathrm{opt})}=\prod_{q=1}^{Q}P_{3(q)}$$

4.Calibration of polarization parameters for a micro-polarizer array detector

The micro-polarizer array detector is used as a polarization detector element. When the extinction ratio is not ideal, the polarization parameters of the micro polarizer array detector should be calibrated, and its influence should be considered in the polarization transfer model of the system. Jones matrix of the micro polarizer array detector $T_{3\left(pol\right)}$ is mainly determined by two parameters, extinction ratio *E* and maximum transmittance *tx*:(8)$$T_{3\left(pol\right)}\left(E,\theta_{\left(pol\right)}\right)=t_{x}\left[\begin{array}{ccc} \cos^{2}\theta_{\left(pol\right)}+\sqrt{E}\sin^{2}\theta_{\left(pol\right)} & \sin\theta_{\left(pol\right)}\cos\theta_{\left(pol\right)}\left(1-\sqrt{E}\right) & 0\\ \sin\theta_{\left(pol\right)}\cos\theta_{\left(pol\right)}\left(1-\sqrt{E}\right) & \sin^{2}\theta_{\left(pol\right)}+\sqrt{E}\cos^{2}\theta_{\left(pol\right)} & 0\\ 0 & 0 & 1 \end{array}\right]$$

Then the three-dimensional global coherency matrix of the polarizer detector $P_{3\left(pol\right)}$ can be expressed as follows:(9)$$P_{3\left(pol\right)}=\left[p_{\left(pol\right)},s_{\left(pol\right)},k_{\left(out\right)}\right]\cdot T_{3\left(pol\right)}\left(E,\theta_{\left(pol\right)}\right)\cdot {\left[p_{\left(pol\right)},s_{\left(pol\right)},k_{\left(out\right)}\right]}^{T}$$

5.The light intensity of different polarization transmission angles

The 3D polarization trace matrix of the entire imaging system $P_{3\left(sys\right)}=P_{3\left(pol\right)}P_{3\left(opt\right)}$. The three-dimensional global coherency matrix of the exiting light $J_{P3\left(\mathrm{out}\right)}$ is obtained as follows.
(10)$$\begin{array}{l} J_{P3\left(\mathrm{out}\right)}=P_{3\left(sys\right)}\cdot J_{P3\left(in\right)}\cdot {P_{3\left(sys\right)}}^{†}\\ P_{3\left(sys\right)}=P_{3\left(pol\right)}P_{3\left(opt\right)} \end{array}$$

Then, the light intensity $I_{\left(out\right)}$ of the incident light is obtained as follows.
(11)$$I_{\left(out\right)}=trace\left(J_{P3\left(out\right)}\right)$$

With the above equation, we can know that the light intensity affected by the polarization error of the optical system has a nonlinear mapping relationship with the incident light intensity, which will be compensated by the depth learning network proposed later.

We add the data obtained from the ZEMAX to the equation and the intensity image is simulated with polarization aberration. The method in detail is as follows:

First, to describe the polarization characteristics of the beam, it is necessary to unify all the beams into one coordinate system, so we transform the local 3D eigen-vibration coordinates into the 3D global coordinates. Suppose that the propagation vector of a beam of light in the global coordinate system O-XYZ is $K={\left[K_{x},K_{y},K_{z}\right]}^{T}$. Its eigen vibration coordinate system is the three-dimensional coordinate system O-X’Y’Z’ obtained by rotating the Z-axis of this global coordinate system in the direction of K (the Z‘ axis is in the same direction as K). The $K_{x}$, $K_{y}$, and $K_{z}$ from the data of the optical system designed by the zemax software are introduced into the equation to obtain the rotation angles $\omega_{y}$ and $\omega_{x}$ to calculate the transformation matrix *R*.
(12)$$\omega_{y}=-(90-\mathrm{arc}\cos(\frac{K_{x}}{\sqrt{K_{x}^{2}+K_{y}^{2}+K_{z}^{2}}})),\;\omega_{x}=a\tan(\frac{K_{y}}{K_{x}})$$
(13)$$R_{x}=\left[\begin{array}{ccc} 1 & 0 & 0\\ 0 & \cos\omega_{x} & \sin\omega_{x}\\ 0 & -\sin\omega_{x} & \cos\omega_{x} \end{array}\right],\;R_{y}=\left[\begin{array}{ccc} \cos\omega_{y} & 0 & -\sin\omega_{x}\\ 0 & 1 & 0\\ \sin\omega_{y} & 0 & \cos\omega_{y} \end{array}\right]$$

The resulting transformation matrix R from the eigen-vibration coordinate system to the global coordinate system is
(14)$$R=R_{x}\cdot R_{y}$$

With this rotation matrix R, the 3D eigen-vibration coordinate matrix can be converted into a 3D global coherency matrix.

Then, we also need to consider the effect of the material-reflecting interface of the DMD on the beam polarization state, and we obtain the reflectivity coefficients $r_{s(q)}$,$r_{p(q)}$ of the *s* and *p* components on the interface *q* from the ZEMAX simulation data. The $T_{3r(q)}$ is expressed as
(15)$$T_{3r(q)}=\left[\begin{array}{ccc} r_{p(q)} & 0 & 0\\ 0 & r_{s(q)} & 0\\ 0 & 0 & 1 \end{array}\right]$$

Finally, by adding the obtained transformation matrix R and $T_{3r(q)}$ to Equations (5) and (6), the polarization intensity images of the four polarization transmission angles after passing through the optical system can be calculated by Equation (11).

### 2.3. Compressed Coding Method

Due to the unique structure of the polarizer array detector, we have improved on the traditional block-compressed coding method. In the process of the conventional block compression coding, each sub-region of the DMD projected to a single pixel of the detector is coded identically. To perceive polarization states, DoFP cameras integrate a micro-polarizer array with four polarization direction (0°, 45°, 90°, 135°) units into the super-pixels of the focal plane array sensor [14]. Due to the unique structure of the polarization detector, we project every 4 × 4 micromirrors on the DMD into a block of 2 × 2 pixels on the detector, where each 1 pixel corresponds to a different DMD encoding, making different encoding patterns for pixels with different polarization transmission angles so that the reconstructed image can be obtained under a lower sampling rate. The conventional micro-polarizer array DoFP camera with 512 × 512 pixels can obtain four intensity images with different polarization transmission angles of 256 × 256 pixels, and the polarization characteristic image with 256 × 256 pixels can be obtained. Our method uses a micro-polarizer DoFP polarization detector with 512 × 512 pixels and a DMD with 1024 × 1024 micromirrors. Multiple intensity images with 256 × 256 pixels are obtained by multiple coded sampling, and the polarization characteristic image with 512 × 512 pixels is gained by the reconstruction network. In this way, the resolution of the polarization characteristic image is doubled.

At each time when the DMD pattern is changed, the light intensity is encoded, and a light intensity image is acquired through the 2 × 2 polarization detector to complete a sampling. Compressed data with different sampling rates can be achieved by controlling the number of detector exposures. Our DMD control system can load 8-bit grayscale patterns by controlling the micromirror dithering. So, we implement the 8-bit quantized versions of random Gaussian measurement matrices.

The 4 × 4 scene is multiplied by the DMD encoding as shown in Figure 4. When the number of samples is *N* = 16, the intensity information pixels collected are 16 × 256 × 256, which is the same amount of information collected by a detector with pixels 1024 × 1024, without compressed sampling, and a 512 × 512 image of polarization information can be reconstructed. Therefore, we designed 16 different 4 × 4 compressed sampling matrixes. We tried compressed sampling imaging at *N* = 12, 8, 4 and 1, i.e., compressed sensing rates CS = 0.75, 0.5, 0.25, and 0.0625, respectively.

### 2.4. Network of This Paper

Unlike the traditional image compression sensing optimal reconstruction algorithms, deep learning need to learn prior knowledge from massive data. It uses a depth neural network to establish the mapping relationship between input and output. The parameters of the network are trained and optimized by a large amount of data. The sequence images collected by the detector in the compression imaging system are input into the trained network, and the reconstructed images can be directly output.

Inspired by ReconNet and ForkNet, we designed a fully connected convolutional super-resolution reconstruction network based on DMD compressed sensing imaging with the polarization aberration correction. Multiple sampled polarized low-resolution images after compression coding are differentially amplified as network inputs, and high-resolution S0, DoLP, and AoP images are used as the network outputs. There is no need to train each polarization transmission angle image separately to obtain the final required key information with a simple network. At the same time, our network effectively avoids the block effect problem of DMD compressed imaging, eliminating the need for additional block effect elimination calculations. Our network has three functions: firstly, it learns the nonlinear mapping relationship between the four angular polarization images and the polarization characteristic images; secondly, it realizes the compressed sampling super-resolution imaging; finally, it compensates the polarization reflection aberration introduced by the DMD and eliminates the block effect. In summary, our goal is to directly reconstruct the high-resolution S0, DoLP, and AoP images required for polarization detection at a lower sampling rate using the proposed convolutional network in combination with DMD compressed imaging techniques.

The architecture of our method is shown in Figure 5. The DoFP polarization detector acquires multiple frames of the compressed coded image, and the size of the frames data is *N* × 256 × 256. Then, the frames are interpolated into *N* × 512 × 512 by bicubic interpolation. The three-dimensional data cube composed of these images is the input to the network. The traditional two-dimensional block network is improved into a three-dimensional network. The polarization imaging principle is combined with the compressed sensing coding method to expand the two-dimensional block information into three-dimensional information. The whole image is used as the network input, which can effectively reduce the block effect brought by the traditional block reconstruction algorithm in blocks and obtain better reconstruction results.

Our network consists of a multilayer convolutional neural network with the convolutional kernel size shown in Figure 5. The step size of the convolution is set to 1, the activation function is ReLu, and we use appropriate zero padding to keep the feature mapping size constant in all layers.

The first layer of the convolutional network has a convolutional kernel size of 1 × 1 and generates 32 feature maps, which mainly extract features in each block. Inspired by Forknet, the second to fourth convolutional layers, respectively, use kernels of size (5 × 5, 3 × 3, 3 × 3) with filters (96, 48, 32), which are used to create a mapping of compressed data to low-resolution features. The previous settings for the fifth to seventh convolutional layers are repeated to create a mapping of low-resolution features to high-resolution features. The last layer, which is also the output of the network, directly outputs the 3D matrix, including the S0, DoLP, and AoP images representing the polarization characteristics. The loss function is the mean square error loss function MSE Loss in the PyTorch model.

### 2.5. Training Data

To train and evaluate the neural network proposed above, we built a dataset containing 100 sets of polarized images. First, we use a high-resolution micro polarizer array camera (2048 × 2448 pixels) to capture different scenes and intercept 1024 × 1024 pixels in the center of the field of view to obtain four 512 × 512 pixel intensity images with polarization transmission angles of 0°, 45°, 90°, and 135°, and s0, AoP, and DoLP images are calculated as the high-resolution ground truth images for the network input. Then, according to the light intensity tracing formula above, the high-resolution ground truth image and the optical system parameters we designed, the projected image with polarization aberration after DMD reflection is simulated. Finally, the projected image is dot-multiplied with the compressed sampling matrix to simulate the compression coding process and obtain multiple low-resolution polarization images acquired by the DoFP detector as the low-resolution image for the network input. In total, 80 sets were used for training data, 10 sets were used as validation data, and the remaining 10 sets were used for testing. To increase the amount of data, we chunked the images in each group. We split the intensity image of size 512 × 512 into smaller pieces of size 64 × 64 and generated more than 5120 patches. In addition, we used flips and rotations (0°, 90°, 180°, and 270°) for data expansion. There are 40,960 patches in total used for training. Figure 6 shows the loss curve on both training and validation sets.

## 3. Results

### 3.1. Simulation

Firstly, the effect of polarization aberration on the intensity image is simulated by the formula in Section 2, and the images with polarization aberration are generated, as shown in Figure 7. We can see that the polarization aberration makes the light intensity value of each polarization transmission angle change significantly.

The dataset established in 2.5 is used to train the network proposed in this paper. We trained the reconstruction network at *N* = (1, 2, 4, 8, 12) and compression rates *cs* = (0.0625, 0.125, 0.25, 0.5, and 0.75), respectively. The PSNR values of the directly reconstructed polarized images are shown in Table 2. It can be seen that the reconstruction network can still reconstruct a high-quality polarization image even at a lower compression rate equal to 0.0625. The reconstructed images with different compression rates of polarization are shown in Figure 8.

To evaluate the reconstruction effect of the algorithm, we compared our algorithm with the compressed sensing algorithm OMP and the deep learning network ReconNet. The reconstruction process is as follows: the inputs of the three algorithms are all low-resolution intensity images of each polarization transmission angle and contain the polarization aberration; OMP and ReconNet are used to reconstruct the high-resolution images of each polarization transmission angle, respectively, whose S0, AoP, and DoLP are calculated by the high-resolution images of each polarization transmission angle after equations (1–3). Our algorithm outputs the polarization data S0, AoP, and DoLP directly without additional calculations. The ground truth images are high-resolution images without polarization aberration. The PSNR value is used to evaluate the reconstruction effect, and the comparison of the reconstruction effect of different algorithms is shown in Figure 9.

Table 3 shows the PSNR values of reconstructed images for different algorithms at a 0.25 sampling rate. It can be seen that the traditional compression sensing algorithm OMP cannot achieve polarization image reconstruction well; ReconNet inevitably brings a mosaic effect, which requires additional correction. In addition, it needs to reconstruct the high-resolution images of four polarization transmission angles first, and then calculate the polarization information image through the formula, which is more complex, and the polarization aberration cannot be compensated. Our algorithm can reconstruct the best image quality at a 0.25 sampling rate in these methods.

### 3.2. Experiment

We want to use this work on the high-resolution polarization imaging in real scenes with the compressed-encoded polarization imaging device we designed. Therefore, we applied the trained network to our designed experimental imaging platform. We have previously analyzed the alignment error of the DMD compression imaging system and pointed out that within the normal tolerance range, the alignment error has no significant impact on the imaging effect [8]. The core components of the imaging system are a DMD from TI, two dual telecentric projection lenses, and a micro-polarizer array camera from LUCID. In the process of real scene data acquisition, the image will also be affected by a variety of environmental uncertainties (noise, vibration, etc.), which can also lead to an inaccurate imaging model. We have done two experiments separately. The first experiment verifies that the DMD reflection system will cause polarization aberration and shows the impact on polarization intensity image; the second experiment uses the built experimental system to acquire real scenes. Through the depth learning network we propose, we can reconstruct the obtained image by super-resolution.

#### 3.2.1. Polarization Aberration Testing of DMD Devices

To verify the influence of DMD on the polarization imaging system, we compared the polarization intensity images acquired by the optical system with and without DMD devices. The integrating sphere is used as a uniform light source, a visible light camera with the same pixel size as the micro polarizer camera is used as a detector, and two rotary polarizers are used as the polarizer and polarizer. First, a rotating polarizer 1 is placed in front of the integrating sphere light source as the polarizer, and a rotating polarizer 2 is placed in front of the visible light camera as the polarizer analyzer, which is set to 0 degrees. Then, the integrating sphere light source is turned on, the polarizer is rotated, and the light intensity values collected by the detector are recorded from 0 degrees to 180 degrees, respectively. Finally, the DMD is added to the imaging system, and all micro mirrors on the DMD are loaded with modulated images with a value of 1, that is, all light incident on the DMD is not coded, but is reflected. Similarly, the polarizer is rotated to record the light intensity reflected by DMD. The two groups of polarization intensity images obtained in the two cases are compared, as shown in Figure 10, and it can be seen that the addition of the DMD has an impact on the polarization light intensity, which needs to be compensated.

#### 3.2.2. Real Scene Polarization Imaging Experiment

To verify the effectiveness of our reconstructed network in real scenarios, we built a two-armed experimental device, as shown in Figure 11. First, the scene is projected on the DMD through the converging lens and the Bi-telecentric objective lens 1. The DMD has a resolution of 1920 × 1080 micromirror elements, each with a size of 10.8 μm, and the micromirrors can be switched at high speed in the ±12° direction to achieve 8-bit grayscale modulation by pulse width modulation. Then, the reflection direction of each micromirror of the DMD is controlled to encode and modulate the scene. Since the DMD micromirrors are flipped diagonally, we rotate the DMD by 45 degrees with an angle of 24° between the two double telecentric lenses for the convenience of mounting. Finally, the encoded scene image is captured by the micro polarization camera after passing through the double telecentric objective 2. The polarization camera is chosen with the Lucid Phoenix camera with IMX250 CMOS, which is also rotated by 45°. We selected a 1024 × 1024 pixel size area on the DMD, corresponding to a 512 × 512 pixel size area on the detector. The whole device is placed on an optical air floating platform to keep it stable.

We photographed the toy vehicle with the experimental device, and the PSNR values of the reconstructed images at different sampling rates are shown in Figure 12. At the sampling rate of 0.0625, the reconstructed images are shown in Figure 13.

Although there is still a large gap between the experimental dataset and the real scene, we still reconstructed a good image quality, which indicates that our algorithm has certain robustness and can be improved even more by increasing the training data of real shooting in the future.

Considering the improvement of some specific application object recognition results with our method, a nylon false leaf is put in the middle of the real leaves, and the polarization information image of it is obtained by our method. It can be seen from Figure 14 that in the AoP image, the grain of the false leaf is obviously different from that of real leaves. From the visual angle, it can be seen that polarization imaging is helpful for the recognition of artificial targets in the natural background. In the future, we will further expand the range of the experiment and give a quantitative analysis of the improvement of different target recognition probabilities.

## 4. Discussion

For polarization imaging, we have established a depth learning network to realize image compression sensing super-resolution reconstruction while compensating polarization aberration. In the reconstruction process, our network establishes the mapping relationship from the low-resolution intensity images with polarization transmission angles to the high-resolution polarization information images, which simplifies the process of first reconstructing the high-resolution intensity images from the original low-resolution images and then calculating the polarization information images by the formula, and also effectively avoids the error superposition brought by multiple super-resolution reconstructions. Compared with traditional methods, the deep compressed sensing algorithm we use not only has a better reconstruction effect and faster reconstruction speed but also has the function of correcting polarization aberration without additional formula and calculation to obtain polarization information images.

The addition of the DMD enables the encoding process in compressed sensing imaging and makes it possible to reconstruct high-resolution images with low sampling rate images. The original polarization intensity image will be affected by polarization aberration caused by DMD, and the images with different polarization transmission angles will be overlapped on the detector. In the experiment, it is found that by using this overlapping, we can reconstruct the polarization information image by a single image with one polarization transmission angle, which is different from the traditional method of reconstructing a polarization information image by the images of four polarization transmission angles. Therefore, the polarization information image with a resolution of 512 × 512 can be reconstructed from a 256 × 256 low-resolution image of a single polarization transmission angle, which means that the sampling rate is 0.0625 (1/16).

At present, the resolution of our method is limited by the DMD resolution. In the visible light band, our imaging method does not significantly improve the resolution. However, in the infrared polarization detection dimension, the resolution can be increased to 2–4 times the original image through appropriate training data sets. Because the high-resolution infrared detector is very expensive, our imaging method can significantly reduce the system’s cost.

We designed a DMD polarization compression imaging optical system using Zemax optical design software and analyzed the influence of the imaging system on the aberration of polarization imaging. The local 3D eigen-vibration coordinate system is rotated and transformed into the 3D global coordinate system, and the influence of the DMD reflection interface on the polarization state is analyzed to obtain a simulated image with polarization aberration of the optical system, which can eliminate the influence of the instrument polarization aberration on the imaging and improve the imaging contrast.

The training data is added with simulated polarization aberration, compressed and sampled to obtain low-resolution images. When reconstructing the image after real DMD reflection imaging with the network trained from the existing dataset, the reconstruction effect has been better than other traditional reconstruction algorithms, but it still needs to be improved. Therefore, for future work, we look forward to using DMD devices to capture large amounts of data for specific needs in order to create more accurate databases for training.

Block-compressed imaging based on deep compressive sensing can achieve fast compression imaging and improve imaging resolution. Polarization imaging can improve detection contrast and suppress flare on the water surface. We can combine the block compressed imaging based on deep compressive sensing and polarization imaging to achieve high-resolution fast polarization imaging. It can be used for sea surface target recognition and tracking.

## 5. Conclusions

In this paper, we propose a polarization imaging method based on DCS by adding a DMD to the optical system to achieve compressed sensing super-resolution imaging. It takes the encoded intensity image of each polarization transmission angle as input and learns the nonlinear mapping between the encoded image and the polarization characteristics directly. The influence of the optical system on the polarization aberration is also analyzed, and the polarization aberration is corrected by network learning while improving the resolution. The network is compared with existing methods and achieves better results in terms of lower sampling rate, quantitative metrics, and visual effects. In addition, the polarization imaging experiments carried out in the paper verify the potential application of the polarization DCS imaging method in the field of high-resolution polarization imaging.

## Figures and Tables

**Figure 1 sensors-22-09676-f001:**
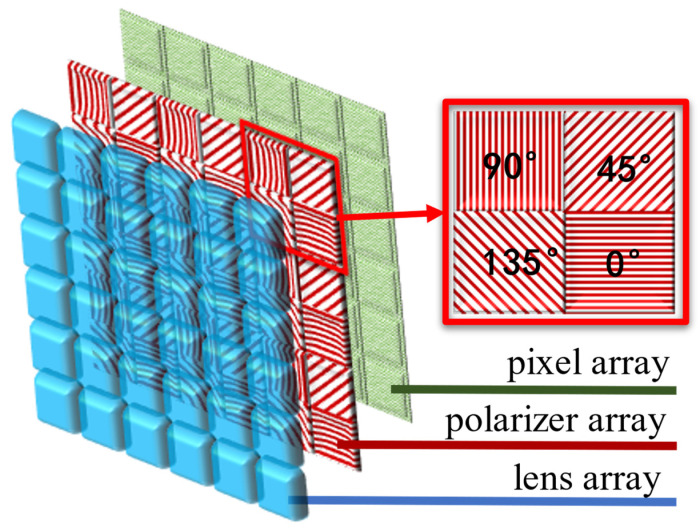
Micro-polarizer array DoFP detector structure.

**Figure 2 sensors-22-09676-f002:**
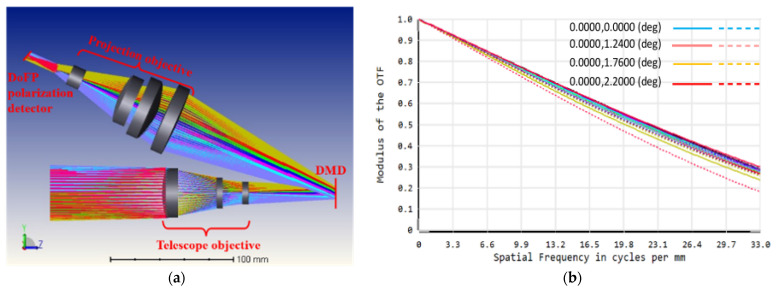
Schematic diagram of DMD-based polarization optical system. (**a**) Structure of the optical system; (**b**) Modulation Transfer Function (MTF) of the optical system.

**Figure 3 sensors-22-09676-f003:**
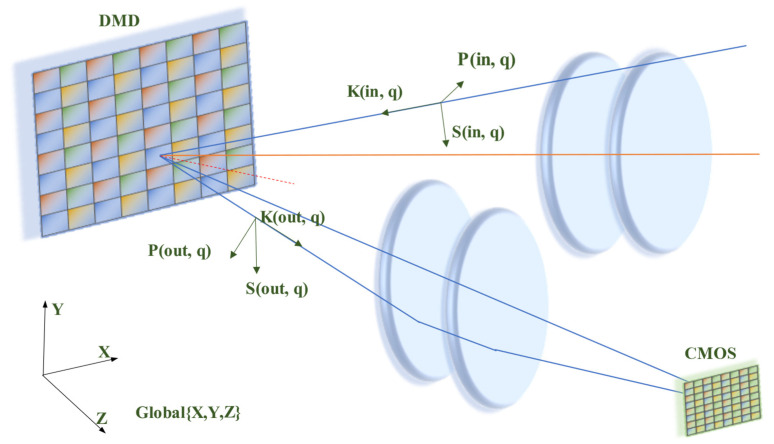
Schematic diagram of the eigen-vibration coordinates (K(in, q), P(in, q), S(in, q)), interface coordinates (K(out, q), P(out, q), S(out, q)), and global coordinates (X,Y,Z).

**Figure 4 sensors-22-09676-f004:**
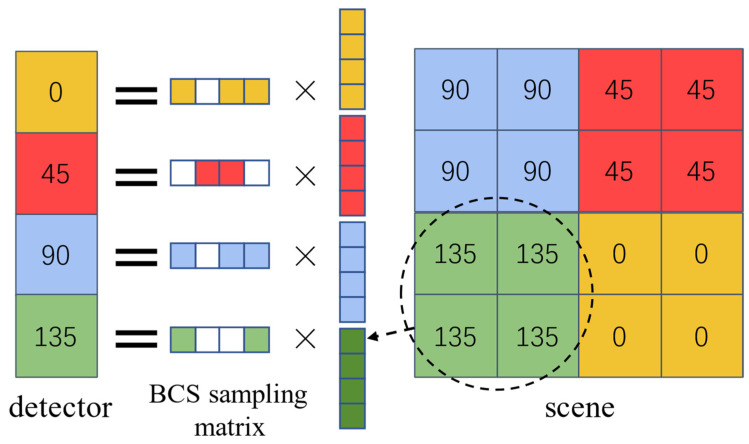
Polarization block compressed sampling by DMD.

**Figure 5 sensors-22-09676-f005:**
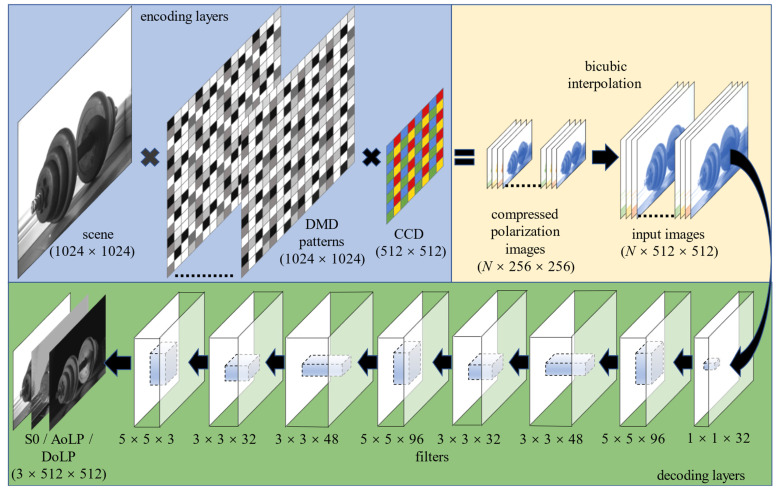
Overall architecture of the compression coding and super-resolution reconstruction network.

**Figure 6 sensors-22-09676-f006:**
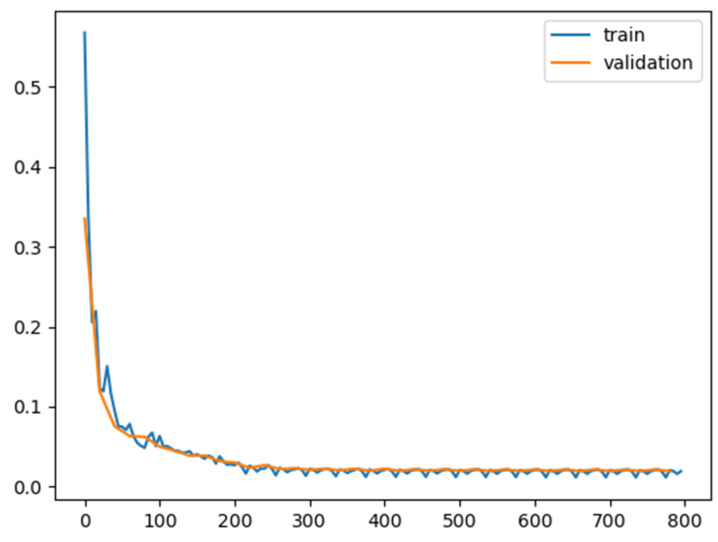
Loss curve on both training and validation sets.

**Figure 7 sensors-22-09676-f007:**
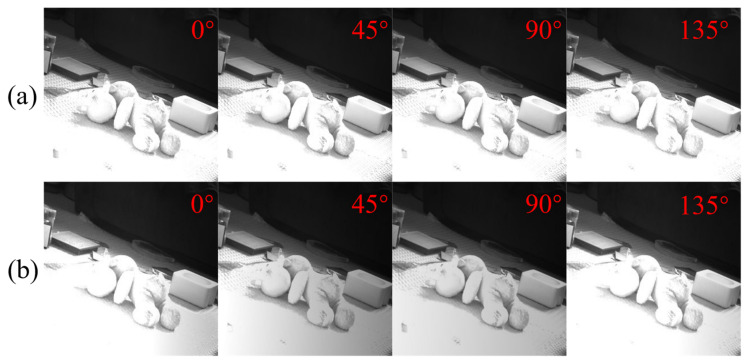
Comparison of intensity images of different polarization transmission angles with and without adding polarization aberration, (**a**) original image, (**b**) simulated image with polarization aberration.

**Figure 8 sensors-22-09676-f008:**
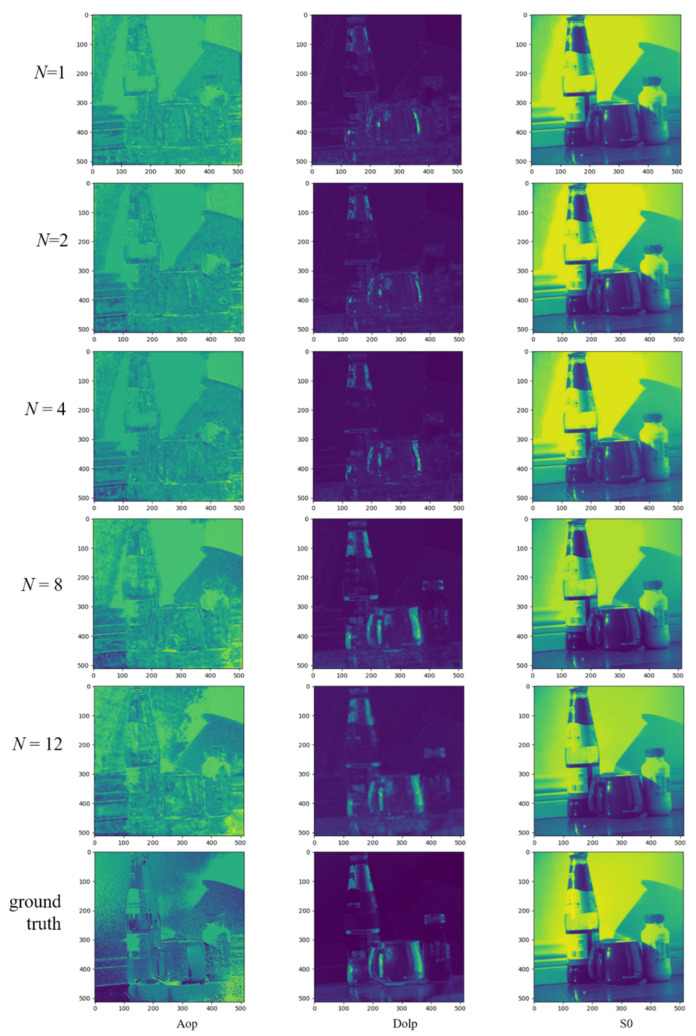
Polarization reconstruction images with different sampling rates.

**Figure 9 sensors-22-09676-f009:**
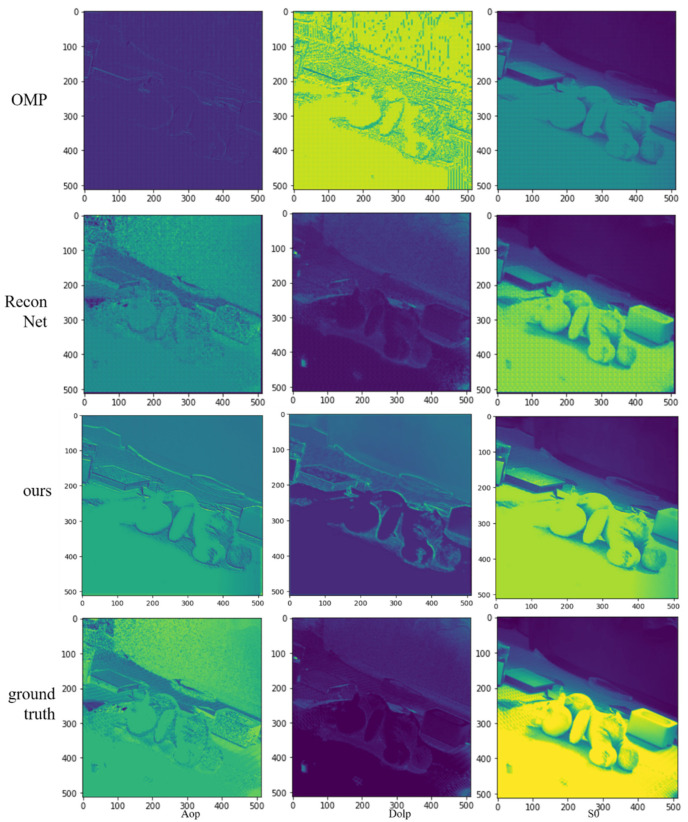
Comparison of reconstruction results of different algorithms.

**Figure 10 sensors-22-09676-f010:**
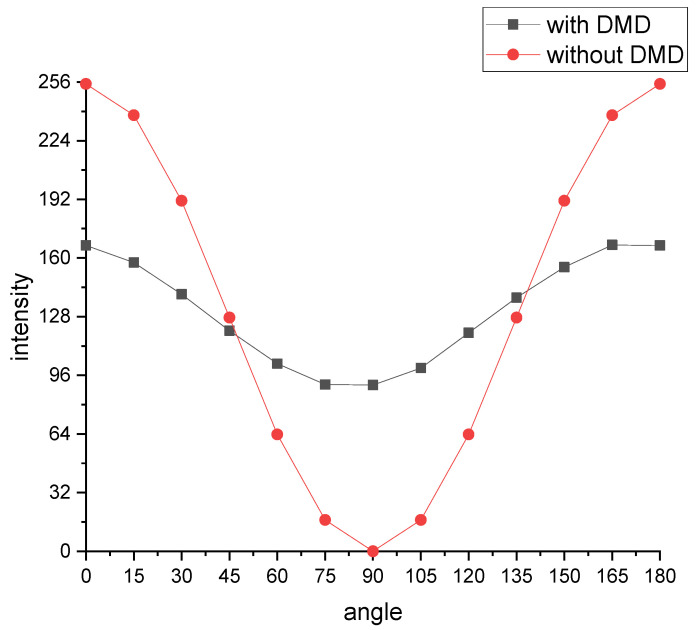
Comparison of polarized light intensity with and without DMD in the optical system.

**Figure 11 sensors-22-09676-f011:**
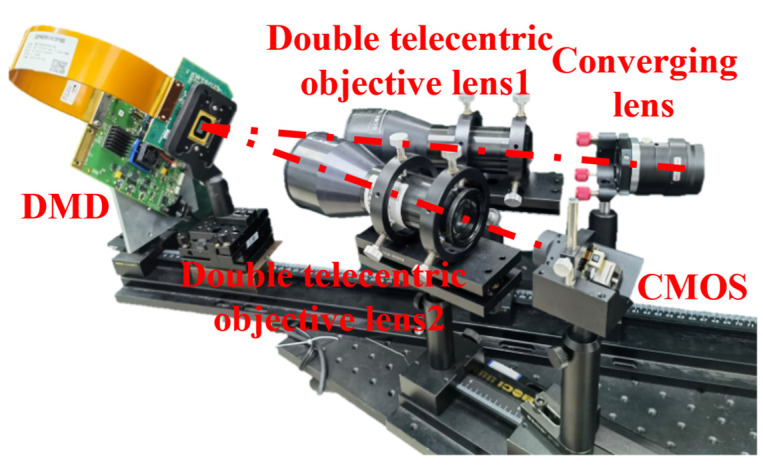
Experimental device.

**Figure 12 sensors-22-09676-f012:**
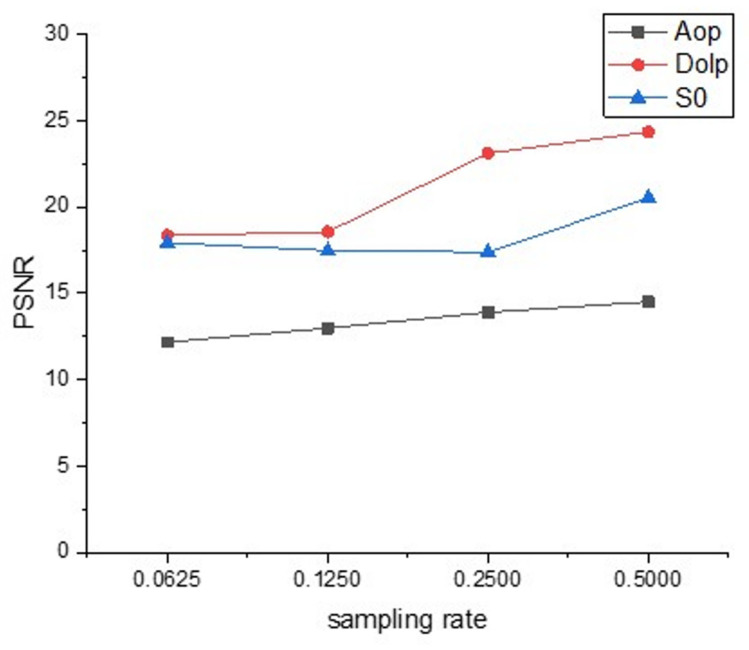
PSNR values of reconstructed images with different sampling rates of real scenes.

**Figure 13 sensors-22-09676-f013:**
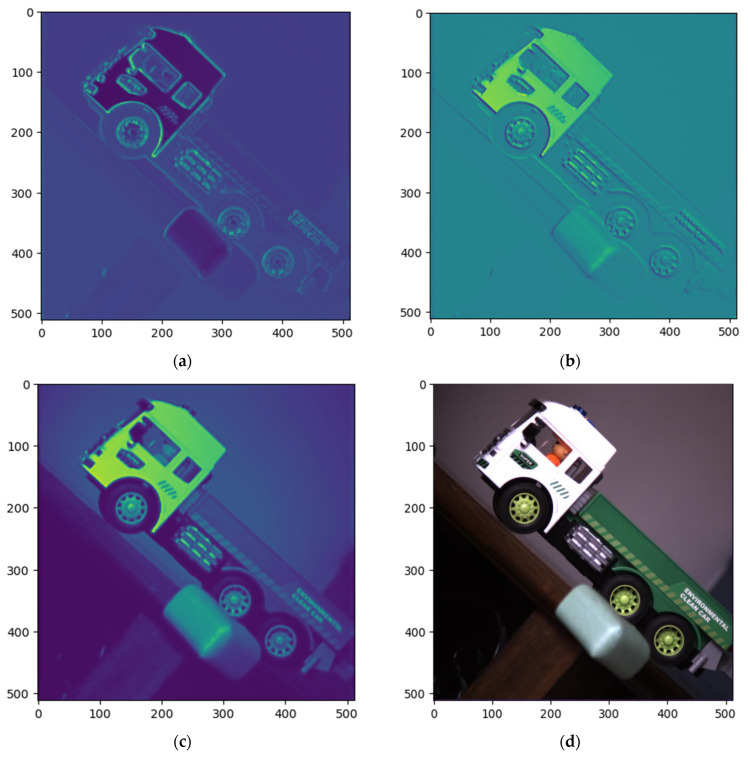
Real scene reconstruction images (**a**) Dolp (**b**) Aop (**c**) S0 (**d**) Grand-truth Visible light.

**Figure 14 sensors-22-09676-f014:**
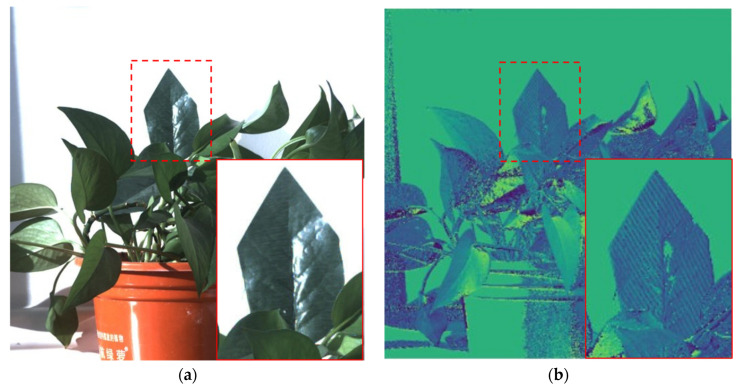
Application of polarization super-resolution imaging in identifying true and false leaves. (**a**) visible light image. (**b**) AoP image after polarization reconstruction.

**Table 1 sensors-22-09676-t001:** System parameters.

Parameter	Value
Wavelength/nm	390–780
FOV(X/Y)	2.2°/2.2°
F/#	1
DMD array size/pixel	1024 × 1024
DMD pixel size/µm	10.8
Detector pixel size/µm	3.45
Detector array size/pixel	512 × 512

**Table 2 sensors-22-09676-t002:** PSNR values of polarization reconstructed images with different compression rates of the test set.

CS Rate	0.0625	0.125	0.25	0.5	0.75
**S0**	30.7730	31.10824	31.8080	35.0894	35.1019
**DoLP**	21.9011	22.4720	23.4041	27.2920	27.7508
**AoP**	14.5859	14.8542	15.0384	15.4558	15.4641

**Table 3 sensors-22-09676-t003:** PSNR values of polarization reconstructed images with different methods of the test set.

	OMP	ReconNet	Ours
**S0**	7.0064	15.3717	29.5582
**Dolp**	1.3242	26.3779	27.1203
**Aop**	5.9693	15.2679	13.6756

## Data Availability

Data underlying the results presented in this Letter are not publicly available at this time but may be obtained from the authors upon reasonable request.

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
