# Peer review of "Polarization Super-Resolution Imaging Method Based on Deep Compressed Sensing"

_sensors, 2022, doi:10.3390/s22249676_

Round 1

Reviewer 2 Report

The research idea on the combination of DoFP and DMD is very enlightening. I still have the following suggestions for the authors to improve:

1)   Please revise Figure 2a, and illustrate the optical system more clearly and mark the devices used.

2)   Please make clear how to control the number of encoded sampling, and then for the case of the number of samples N=16, what are the compressed sampling matrixes corresponding to different encoded sampling?

3)   The DoFP polarization detector acquires multiple frames of the compressed coded image, the size of the frames data is Nx256x256. Then the frames are interpolated into Nx512x512.” What interpolation method is used here?

4)   I cannot see Figure 11, please reinsert the image.

5)    Except for PSNR values of polarization reconstructed images, which other indicators can the authors use to show the improvement of image recognition results by this method for some specific application objects.

Round 2

Reviewer 1 Report

Authors have addressed the comments satisfactorily.  This research paper may be accepted in the present form.